# HOW TO KEEP COOL WHILE TRAINING

## ABSTRACT

Modern neural networks used for classification are notoriously prone to overly confident predictions. With multiple calibration methods proposed so far, there has been noteworthy progress in addressing overconfidence issues. However, to the best of our knowledge, prior methods have exclusively focused on those factors that affect calibration, leaving open the question of how (mis)calibration circles back to negatively impact network training. Aiming to better understand such dependencies, we propose a temperature-based *Cooling* method to calibrate classification neural networks during training. *Cooling* results in better gradient scaling and reduces the need for a learning rate schedule. We investigate different variants of Cooling, with the simplest, *last layer Cooling*, being also the best-performing one, improving network performance for a range of datasets, network architectures, and hyperparameter settings.

## 1 INTRODUCTION

Training neural networks can be a challenging task, with optimal performance depending on the right setting of hyperparameters. For this reason, finding a suitable network configuration can often take multiple costly training runs with varying parameters of the learning rate schedule, the optimizer and the batch size. Apart from standard learning rate schedules like piecewise constant schedules and exponential decay schedules, there has been activate research in developing better schedules: Among the most prominent of these are learning rate warmup (Goyal et al., 2017; He et al., 2016a) and cosine decay (Loshchilov & Hutter, 2017) schedules.

Complementary to these challenges, (Guo et al., 2017) found that modern convolutional classification networks are often poorly calibrated, leading to overly confident predictions. They investigated multiple methods to improve calibration, with a simple *temperature scaling* method performing best: the network's output logits are multiplied by a temperature parameter, optimised on a validation dataset after training. Importantly, this leaves the maximal value and therefore the predicted class label unchanged since all the logits are multiplied by the same temperature value.

Since then, multiple papers (Kull et al., 2019; Kumar et al., 2019; 2018; Müller et al., 2019; Gupta et al., 2021) proposed methods aiming to even better calibrated networks. More recently, (Desai & Durrett, 2020; Minderer et al., 2021) investigated the calibration of state-of-the-art non-convolutional Transformer networks (Vaswani et al., 2017; Dosovitskiy et al., 2021) and MLP-Mixers (Tolstikhin et al., 2021). They concluded that such architectures may have benefits, with further work needed to fully understand the factors contributing to calibration.

Despite initially leaving the accuracy unchanged, we have noticed that temperature scaling can have an intriguing effect as training continues: scaling the output logits results in a change in the cross-entropy loss, which in turn leads to scaled gradient updates and subsequently new parameter values. During training, this can lead to a significant increase in accuracy. To the best of our knowledge, temperature scaling has until now only been applied *post hoc* after completing network training. However, our investigation shows that networks become gradually overconfident during training (they *overheat*), which seems to have a detrimental effect on learning. This has motivated us to modify the original temperature scaling and propose a *Cooling* method to calibrate neural networks *during* training.

**Our Contributions**

- A *Cooling* method for calibrating classification neural networks during training. We propose two basic variants called *last layer Cooling* and *distributed Cooling*, and one hybrid variant called *periodically redistributed Cooling*.

- A mathematical analysis of the effect of Cooling on the network gradients, with a comparison of different Cooling variants.

- An empirical investigation of the effects of Cooling on a range of metrics, including network weights, gradients, output logits and the ECE (expected calibration error) calibration measure.

- A broad set of experiments for different tasks (image classification and semantic segmentation), datasets and network architectures. We also include an extensive ablation study, involving different activation functions, optimizers, and hyperparameters such as the learning rate schedule, the Cooling factor and the use of weight decay and data augmentation. Our experiments indicate an interplay between the learning rate and calibration during training. Importantly, if well-calibrated, networks can train well without the use of a learning rate schedule.

## 2  BACKGROUND AND NOTATION

Let $f_\theta : \mathbb{R}^d \to \mathbb{R}^s$ denote the function of a classification neural network with parameters $\theta$, mapping a $d$-dimensional input (in our case an image) $\mathbf{x}$ to an $s$-dimensional logits vector $\mathbf{z} = f_\theta(\mathbf{x})$. During training, each input $\mathbf{x}$ comes with a class-probability or label vector $\mathbf{y}$, denoting probabilities of $\mathbf{x}$ belonging each of $s$ classes. This is usually (but not necessarily) a one-hot vector corresponding to a so-called ground-truth class label, $i^*$.

In its simplest variant, we suppose the network consists of $L$ affine (dense or convolutional) layers, each followed by a non-linear activation function. For the $i^{\text{th}}$ layer ($1 \leq i \leq L$) this gives an expression of the form $\mathbf{x}_i = \rho(\mathbf{p}_i) = \rho(\mathbf{W}_i \mathbf{x}_{i-1} + \mathbf{b}_i)$ with weight matrices $\mathbf{W}_i$, bias vectors $\mathbf{b}_i$, non-linearities $\rho$, pre-activation values $\mathbf{p}_i$ and layer inputs and outputs $\mathbf{x}_{i-1}$ and output $\mathbf{x}_i$, respectively. (More generally, our method can be applied to any neural network, involving arbitrary functions and layers like e.g. attention, batch normalization and skip connections.) The output logits are then passed through the softmax function $\sigma$ which results in a vector $\widehat{\mathbf{y}} = \sigma(\mathbf{z})$ of class probabilities. The classification network is trained to minimize the categorical cross-entropy loss function

$$L(\mathbf{z}) = H(\mathbf{y}, \sigma(\mathbf{z})) = -\sum_{i=1}^{s} y_i \log(\widehat{y}_i) . \tag{2.1}$$

We say that a network is *well-calibrated* if the output values $\widehat{\mathbf{y}}$ can be interpreted as true probabilities. Intuitively, if a network makes 100 predictions with 90% confidence, we would expect that 90% are correctly classified. (Guo et al., 2017) observed that convolutional neural networks tend to display over-confidence in their results, in that $\widehat{y}_{i^*}$ gives an over-estimate of the probability that $\lambda_i$ is the correct label. Thus the networks are badly calibrated, which we metaphorically express by saying that the networks become *overheated*.

(Guo et al., 2017) found that simply multiplying the pre-softmax *logits* $z_i$ by a factor $\tau$ does an excellent task of improving the network's calibration. Thus, the task of correcting the calibration of the network is to find a constant $\tau$ to correct its output, so that it becomes

$$\mathbf{y} = \sigma(\tau \mathbf{z}) = \sigma(\tau f_\theta(\mathbf{x})) . \tag{2.2}$$

The optimal $\tau$ is found by minimizing the log-likelihood cost function on a small *calibration set*, held back from the training data. This operation is carried out when the network is fully trained. Usually, one finds that the optimum value is $\tau < 1$. This process is known as *temperature scaling*.

We refer to (Guo et al., 2017) for a more detailed introduction to network calibration.

## 3 Method: Cooling While Training

### 3.1 Fundamentals

When the network overheats, the predicted values $\widehat{y}_i$ become too close to $0$ or $1$. This can cause problems with gradients becoming large. We hypothesise therefore that keeping the network at the correct temperature during training can lead to improved convergence. Our proposed operation is to periodically correct the network, by multiplying the logits $z_i$ by the optimal temperature correcting constant. There are multiple ways to implement Cooling, the most basic being *last layer Cooling*:

**Definition 3.1.** A network performs *last layer Cooling* if before the softmax function there is a final scaling layer, multiplying the network output logits $\mathbf{z}$ by a constant scalar $\tau > 0$. This value $\tau$ is not modified during the batch updates of the gradients, but is corrected using a held-out validation set at the end of the Cooling period.

**Cooling factor.** We investigate the effect of taking the optimal temperature parameter $\tau$ to some power $\kappa$, which we call the *Cooling factor*. Let us assume that $\tau < 1$ (which is mostly the case). Then for $\kappa > 1$, multiplying by $\tau^\kappa$ results in smaller logits, a scenario which we call *overcooling*. Conversely, $\kappa < 1$ produces larger logits, resulting in an *undercooling* of the network. We note that as $\kappa \to 0$, we approach standard network training without Cooling. As we show in the experiments, using a suitable value for $\kappa$ can have positive effects on the training stability and performance.

**Cooling period.** We call the periodic time interval after which we perform temperature scaling the *Cooling period*. Typically, we let the Cooling period be equal to one epoch of training. We trained the VGG-style network described in §4.2 on CIFAR10, using an Nvidia GeForce GTX Titan Z GPU and a 16-core Intel Xeon CPU E5-2640 v3. The actual training takes approximately 56 seconds per epoch, whereas Cooling takes approximately 1.05 seconds. Hence, one run of Cooling takes approximately 1.9% of the time of one training epoch. We experimented on CIFAR10 with a Cooling period of one batch, but found no additional performance benefits. Thus, it seems that Cooling once per epoch is sufficient.

### 3.2 Distributed Cooling

Instead of waiting to perform scaling after the final layer, it is possible to redistribute the temperature correction across the network, by scaling layers other than the last.

Suppose the optimal temperature is $\tau \in \mathbb{R}^+$. When we redistribute the temperature across the network, we would like the temperature correction to gradually take effect. More precisely, we want to ensure that (1) each layer multiplies its input by an additional factor of $\beta = \tau^{1/L}$, so that the output of the $i^{\text{th}}$ layer is now $\mathbf{x}_i' = \beta^i \mathbf{x}_i$. Moreover, (2) the inputs to the non-linearities $\rho$ have to be the same as before scaling because we would otherwise change the network output in a non-linear manner. Finally, (3) the output logits $\mathbf{z}$ should be multiplied by $\beta^L = \tau$.

**Definition 3.2.** Let the notation be as in § 2. Let $\beta = \tau^{1/L}$. A network performs *distributed Cooling* if after each Cooling period,

1. the weight matrix $\mathbf{W}_i$ is multiplied by $\beta$, resulting in a new matrix $\mathbf{W}_i' := \beta \mathbf{W}_i$;

2. the bias vector $\mathbf{b}_i$ is multiplied by $\beta^i$, resulting in a new vector $\mathbf{b}_i' := \beta^i \mathbf{b}_i$;

3. the activation $\rho$ is changed to $\rho_{\beta^i}$, defined by $\rho_{\beta^i}(x) = \beta^i \rho(\beta^{-i} x)$.

**Lemma 3.3.** *If a network performs distributed Cooling, then, compared to no Cooling,*

1. *the output of the $i^{th}$ layer is $\mathbf{x}_i' = \beta^i \mathbf{x}_i$, for $1 \leq i \leq L - 1$;*

2. *the input to each of the non-linearities is left unchanged;*

3. *the output logits are scaled by a factor of $\tau$: $\mathbf{z}' = \tau \mathbf{z}$.*

*Proof.* We give the proof in in Appendix A. $\qquad\square$

---

**Algorithm 3.1** Periodically Redistributed Cooling (simplified)

---

**Inputs**:     Neural network $f_\theta$ with $L$ linear layers $(\mathbf{W}_i, \mathbf{b}_i)$
                Final Cooling layer $s_\tau$
                Validation data set $\mathcal{X}_{\text{val}}$
                Cooling factor $\kappa$
                Maximal temperature $\tau_{\text{max}}$; reset temperature $\tau_{\text{re}}$
**Output**:    Calibrated Network $f_{\theta'}$
**After each Cooling period:**

 1:  $\tau \leftarrow$ temperature minimising cross entropy on $\mathcal{X}_{\text{val}}$
 2:  $\tau \leftarrow \tau^\kappa$                                                    $\triangleright$ over- or undercooling
 3:  **if** $\tau > \tau_{\text{max}}$ **then**
 4:       excess $\leftarrow \tau/\tau_{\text{re}}$
 5:       $\tau \leftarrow \tau_{\text{re}}$
 6:       kernel scale $\leftarrow \sqrt[L]{\text{excess}}$
 7:       bias scale $\leftarrow 1$
 8:       **for** $i = 1$ to $L$ **do**
 9:            bias scale $\leftarrow$ bias scale $*$ kernel scale
10:           $\mathbf{W}_i \leftarrow \mathbf{W}_i/\text{kernel scale}$
11:           $\mathbf{b}_i \leftarrow \mathbf{b}_i/\text{bias scale}$
12:  Update $s_\tau$ using new $\tau$

---

It may be observed that if $\rho$ is the ReLU function, then $\rho_{\beta^i} = \rho$, so the activation layer is not changed. (More generally, this holds for piecewise-linear ReLU variants such as CReLU, Leaky ReLU and PRELU.) Note also that $\tau$ is usually less than 1, indicating overheating of the network, and hence multiplications by $\beta$ and $\beta^i$ result in a decrease in the values of the parameters $\mathbf{W}_i$ and $\mathbf{b}_i$.

When performing last layer Cooling, we have observed that the network can correct for this by overheating even more. This means that the required temperature correction $\tau$ becomes smaller and smaller, towards zero. At the same time, this means that the parameters of the layers (for instance $\mathbf{W}$ and $\mathbf{b}$ in affine layers) become larger and larger, eventually causing numerical overflow. A possible solution to this problem is to keep track of the overheating parameter $\tau$, and when it becomes too small, redistribute the temperature correction over all layers:

**Definition 3.4.** A network performs *periodically redistributed Cooling* if for some $\tau_{\text{max}}, \tau_{\text{re}} > 0$,

- it performs last layer Cooling as long as $\tau < \tau_{\text{max}}$ (i.e. the optimal temperature is less than a specified maximal temperature);

- it redistributes the excess temperature $\tau/\tau_{\text{re}}$ across the layers as in Definition 3.2 if $\tau > \tau_{\text{max}}$.

The values $\tau_{\text{max}}$ and $\tau_{re}$ are manually specified. In our experiments, we used $\tau_{\text{max}} = \tau_{\text{re}} = 100$ for periodically redistributed Cooling, but other values can also be considered.

**Summary.** Algorithm 3.1 gives the pseudocode of our proposed Cooling method. The algorithm shows periodically redistributed Cooling, since it is the most general case. We can recover last layer Cooling and distributed Cooling as special cases with $(\tau_{\text{max}}, \tau_{\text{re}}) = (\infty, \infty)$ and $(\tau_{\text{max}}, \tau_{\text{re}}) = (0, 1)$, respectively. The algorithm displayed above is simplified for the sake of clarity: it does not take heed of layers like attention and batch normalization and excludes skip connections. These layers are straightforward to address in a general implementation of the Cooling method.

### 3.3    Effects on gradient values

We now perform an analysis of the effect Cooling on the network gradients.

### 3.3.1 LAST LAYER COOLING

**Proposition 3.5** (Gradients of last layer Cooling). *Let the notation be as in § 2. If $C = L(\sigma(\tau \mathbf{z}))$ where $\sigma$ is the softmax function and $L$ is the cross-entropy loss, the derivative with respect to any network parameter $w \in \theta$ is given by*

$$\frac{\partial C}{\partial w} = \left\langle \sigma(\tau \mathbf{z}) - \mathbf{y}, \frac{\partial \tau \mathbf{z}}{\partial w} \right\rangle = \tau \left\langle \sigma(\tau \mathbf{z}) - \mathbf{y}, \frac{\partial \mathbf{z}}{\partial w} \right\rangle, \tag{3.1}$$

*expressed as the inner-product of two vectors.*

*Proof.* When $\tau = 1$, Equation 3.1 is a known result of a simple computation. The general case is an application of the chain rule. □

**Interpretation.** The difference $\epsilon = \sigma(\mathbf{z}) - \mathbf{y}$ may be termed the *residual*, namely the difference between the ground-truth label probabilities $\mathbf{y}$ and the label probabilities $\sigma(\mathbf{z})$ computed by the network. The derivative $\partial C / \partial w$ is then the inner product of this residual with the vector $\partial \mathbf{z} / \partial w$.

We can use this formula to analyze the effect of temperature-scaling by $\tau$. Suppose $\tau = 1$, so there is no heating correction. As is known, in this case probabilities tend to be overestimated, so that $\sigma(\mathbf{z})$ approaches $\mathbf{y}$. All values of $\sigma(\mathbf{z})$ other than the ground truth become very small, meaning that values of $\partial z_i / \partial w$ are multiplied by small values, and so are ultimately ignored, with harmful effects on convergence.

Setting $\tau < 1$ results in a more evenly distributed (less peaked) vector $\sigma(\tau \mathbf{z})$, meaning that all values of $z_i$ and $\partial z_i / \partial w$ have an effect on the gradient.

### 3.3.2 DISTRIBUTED COOLING

Now, we consider what happens to gradients when distributed Cooling is applied. We simplify the analysis by thinking of scaling occurring in two steps. First global temperature scaling is applied by modifying the final layer so that its output is multiplied by $\tau$. Subsequently, distributed scaling is applied to all layers resulting in the output of the $i$-th layer being multiplied by $\beta_i$, in a way that the network output is unchanged. The effect of the final-layer scaling on gradients was addressed earlier. Now we concentrate on the effect of distributed scaling on gradients in the network.

Consider a network with $N$ layers, labelled $0$ to $N - 1$, let $\mathbf{x}_i$ be the input to the $i$-th layer (which is also the output of the $i - 1$-th layer), and $\mathbf{x}_N$ the output of the last layer. Let another network have layer inputs denoted by $\mathbf{x}_i'$.

**Definition 3.6.** We will say that two networks are *scale-equivalent* if for inputs $\mathbf{x}_0 = \mathbf{x}_0'$ there are constants $\beta_i$ with $\beta_0 = \beta_N = 1$ such that $\mathbf{x}_i' = \beta_i \mathbf{x}_i$. Evidently, for the same input $\mathbf{x}_0 = \mathbf{x}_0'$, the outputs $\mathbf{x}_N = \mathbf{x}_N'$ are the same, since $\beta_N = 1$.

It will be observed, however that the gradients of the parameters of these networks will be different. Let the first (unprimed) network be represented by $x_{i+1} = \rho_i(\mathbf{W}_i x_i + \mathbf{b}_i)$ where $\rho_i$ is an activation function, possibly different for each $i$. Then, given numbers $\beta_i$ with $\beta_0 = \beta_N = 1$, an equivalent network is given by $x_{i+1}' = \rho_i'(\mathbf{W}_i' x_i' + \mathbf{b}_i')$ where

$$\mathbf{W}_i' = \beta_{i+1} \beta_i^{-1} \mathbf{W}_i \quad \text{and} \quad \mathbf{b}_i' = \beta_{i+1} \mathbf{b}_i \quad \text{and} \quad \rho_i' = \rho_{\beta_{i+1}} \tag{3.2}$$

and $\rho_\beta$ is a modified activation function given by $\rho_\beta(\mathbf{x}) = \beta \rho(\beta^{-1} \mathbf{x})$. (It should be noted that if $\rho$ is a ReLU activation, then $\rho = \rho_\beta$.) Then, $x_i' = \beta_i x_i$ for all $i$, as required.

Thus, with $T_i$ representing the transformation $\mathbf{x}_i \mapsto \rho_i(\mathbf{W}_i \mathbf{x}_i + \mathbf{b}_i) = \mathbf{x}_{i+1}$, (and similarly $T'$) we compare the two networks:

$$\sigma \circ T_{N-1} \circ T_{N-2} \circ \ldots \circ T_0(\mathbf{x}_0) \quad \text{and} \quad \sigma \circ T_{N-1}' \circ T_{N-2}' \circ \ldots \circ T_0'(\mathbf{x}_0) \,,$$

where $\sigma$ represents the final softmax layer.

It is evident that these two networks carry out the same operation. However, it will be shown that if optimized using a gradient-descent based method, the update of their parameters will be different,

and the trajectory of the parameters in the path towards the optimum during training will be quite different.

Let $\mathbf{W}_{i,jk}$ be one of the entries of $\mathbf{W}_i$ and $b_{i,j}$ be one of the parameters of $\mathbf{b}_i$. Similarly, let $\mathbf{W}'_{i,jk}$ and $b'_{i,j}$ be the corresponding parameter of the primed (distributively scaled) network. The following will be shown:

**Theorem 3.7.** *Let $C = L(\mathbf{z})$ where $\mathbf{z} = \sigma \circ f_\theta(\mathbf{x}_0) = \sigma \circ f'_{\theta'}(\mathbf{x}_0)$ for scale equivalent networks $f_\theta$ and $f'_{\theta'}$. Functions $\sigma$ and $L$ are softmax and loss functions. Let $\mathbf{W}_{i,jk}$ be the $(j,k)$-th entry of parameter matrix $\mathbf{W}_i$ and $b_{i,j}$ the $j$-th entry of parameter vector $\mathbf{b}_i$. Then*

$$\frac{\partial C}{\partial \mathbf{W}'_{i,jk}} = \frac{\beta_i}{\beta_{i+1}} \frac{\partial C}{\partial \mathbf{W}_{i,jk}} \quad and \quad \frac{\partial C}{\partial b'_{i,j}} = \frac{1}{\beta_{i+1}} \frac{\partial C}{\partial b_{i,j}} \tag{3.3}$$

*Proof.* Define $\mathbf{p}_{i+1} = \mathbf{W}_i \mathbf{x}_i + \mathbf{b}_i$, and $\mathbf{x}_{i+1} = \rho_i(\mathbf{p}_{i+1})$, and similarly primed quantities. We see that $\mathbf{x}'_i = \beta_i \mathbf{x}_i$ and $\mathbf{p}'_i = \beta_i \mathbf{p}_i$ for all $i$. Let $f_{i+1}$ be the mapping defined by

$$C = f_{i+1}(\mathbf{p}_{i+1}) = L \circ \sigma \circ T_{N-1} \circ T_{N-2} \circ \ldots \circ T_{i+1} \circ \rho_i(\mathbf{p}_{i+1})$$

namely, the part of the network "downstream" from $\mathbf{p}_{i+1}$ (including the activation function $\rho_i$ in the $i$-th layer, and the softmax and loss functions). Function $f'_{i+1}$ is similarly defined for the primed network.

We apply the chain rule:

$$\frac{\partial C}{\partial \mathbf{W}_{i,jk}} = \frac{\partial C}{\partial \mathbf{p}_{i+1}} \frac{\partial \mathbf{p}_{i+1}}{\partial \mathbf{W}_{i,jk}}$$

A similar formula holds for the primed case.

Now, since $C = f_{i+1}(\mathbf{p}_{i+1}) = f'_{i+1}(\mathbf{p}'_{i+1}) = f'_{i+1}(\beta_{i+1}\mathbf{p}_{i+1})$ we see

$$\frac{\partial C}{\partial \mathbf{p}'_{i+1}} = \boxed{\beta_{i+1}^{-1}} \frac{\partial C}{\partial \mathbf{p}_{i+1}} \tag{3.4}$$

Next, we compare $\partial \mathbf{p}_{i+1}/\partial \mathbf{W}_{i,jk}$ and $\partial \mathbf{p}'_{i+1}/\partial \mathbf{W}'_{i,jk}$.

Let $\mathbf{E}_{jk}$ be the matrix with an entry 1 in position $(j,k)$ and 0 elsewhere. Then $\partial \mathbf{p}_{i+1}/\partial \mathbf{W}_{i,jk} = \mathbf{E}_{jk}\mathbf{x}_i$. On the other hand,

$$\partial \mathbf{p}'_{i+1}/\partial \mathbf{W}'_{i,jk} = \mathbf{E}_{jk}\mathbf{x}'_i = \mathbf{E}_{jk}(\beta_i \mathbf{x}_i) = \boxed{\beta_i}\, \partial \mathbf{p}_{i+1}/\partial \mathbf{W}_{i,jk}$$

Putting this together with equation equation 3.4 we see that $\partial C/\partial \mathbf{W}'_{i,jk} = (\beta_i/\beta_{i+1})\, \partial C/\partial \mathbf{W}_{i,jk}$ as required.

In the case where $b_{i,j}$ is an entry of $\mathbf{b}_i$, we see that $\partial \mathbf{p}'_{i+1}/\partial b'_{i,j} = \partial \mathbf{p}_{i+1}/\partial b_{i,j}$ so,

$$\partial C/\partial b'_{i,j} = (1/\beta_{i+1})\, \partial C/\partial b_{i,j}\ .$$

$\square$

**Relative gradients.** Since a small change to a small parameter is more important to the same change to a large parameter, it is perhaps more important to determine the ratio $(\partial C/\partial \theta)/\theta$, which determines by what ratio a parameter is changed during gradient update. This gives:

$$\frac{\partial C/\partial \mathbf{W}'_{i,jk}}{\mathbf{W}'_{i,jk}} = \frac{\beta_i^2}{\beta_{i+1}^2} \frac{\partial C/\partial \mathbf{W}_{i,jk}}{\mathbf{W}_{i,jk}}$$

$$\frac{\partial C/\partial b'_{i,j}}{b'_{i,j}} = \frac{1}{\beta_{i+1}^2} \frac{\partial C/\partial b_{i,j}}{b_{i,j}}$$

**Interpretation.** The effect of distributed scaling is to individually change the relative effect of gradients over the network. In particular, if $\beta_i = \tau^{i/N}$, for $i = 0, \ldots, N-1$ and $\beta_N = 1$, distributing the scale evenly across the network, with $\tau < 1$, then the effect is to modify the gradients and relative gradients across the network. This may have the effect of mitigating the effect of gradient vanishing. Distributed scaling can be used to control the magnitudes of the output $\mathbf{x}_i$ at each level.

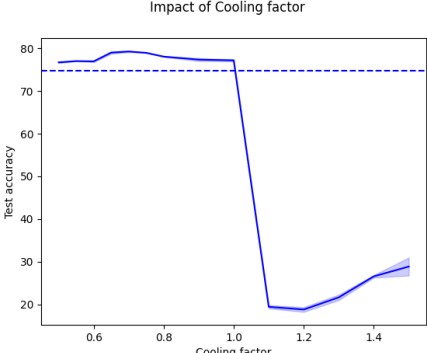 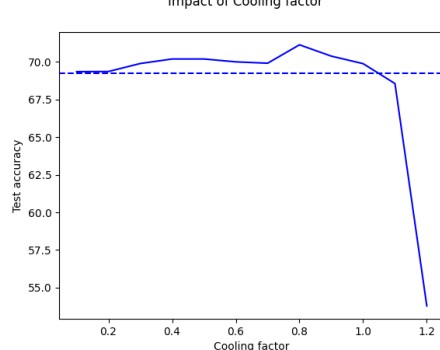

Figure 1: Plots showing the test accuracy of networks trained with last layer Cooling and various Cooling factors. Left: VGG network trained on CIFAR10. Right: ResNet50 network trained on CIFAR100. The non-Cooling baselines are shown as dashed. All Cooling factors $\kappa \leq 1$ outperform the baseline, the best one by $4.6\%$. Training diverges for Cooling factors $\kappa > 1$. (All means and standard deviations are computed over three runs.)

## 4 EXPERIMENTS

### 4.1 GENERAL SETUP

We explore the use of Cooling on two image classification datasets and one semantic segmentation dataset. We use 99% of the CIFAR "training sets" (corresponding to 50,000 images in total) for training and 1% as a validation set to optimise the temperature $\tau$ on. We train our networks using either the SGD optimizer with a momentum of 0.9 or the Adam optimizer (Kingma & Ba, 2015) with $\epsilon = 0.1, \beta_1 = 0.9, \beta_2 = 0.999$. All networks are trained using the TensorFlow (Abadi et al., 2015) framework.

### 4.2 IMAGE CLASSIFICATION: CIFAR10

**Setup.** We train a small VGG-style network (Simonyan & Zisserman, 2014) on the CIFAR10 dataset (Krizhevsky, 2009). The network consists of a sequence of 6 convolutional layers of filter size $3 \times 3$ with $32, 32, 64, 64, 128$ and $128$ channels, respectively, followed by two dense layers with 128 and 10 output nodes, respectively. In total, the network has approximately 620,000 trainable parameters. The hidden layers either use the ReLU (Fukushima, 1980; Nair & Hinton, 2010) or the CReLU (Shang et al., 2016) activation. When we use learning rate warmup, we linearly increase the learning rate from 0 to 0.01 over 2 epochs. When we do not use warmup, we directly start with a learning rate of 0.01. In our learning rate schedule ablations, we experiment with (1) no schedule, (2) a piecewise linear schedule which drops by a factor of 0.1 after 30 and 40 epochs, (3) a schedule with linear decay from the initial rate to 0, a (4) schedule with exponential decay with a total drop by either a factor of 0.01 ("slow") or 0.001 ("fast") and (5) a cosine decay schedule (Loshchilov & Hutter, 2017). We train the network for 50 epochs. We use a batch size of 64.

**Results.** As shown in Table 1, Cooling can have a significantly positive impact on network performance. We see a stark difference between various LR schedules. Whereas smooth schedules (where the learning rate changes after each batch) hardly benefit from Cooling, there is a noticeable benefit for piecewise constant schedules and a drastic improvement when no schedule is employed. Starting from the lowest performance at 74.7% test accuracy, last layer Cooling increases the test accuracy by 4.6%.

Figure 1 (left) shows an ablation of last layer Cooling, involving various Cooling factors $\kappa$. Last layer Cooling shows little sensitivity to the Cooling factor, as long as $\kappa \leq 1.0$. Values greater than 1.0 lead to divergence. On the other hand, all values of $\kappa \leq 1.0$ produce networks outperforming the non-Cooling baseline.

| LR Schedule | No Cooling | Distributed | Periodic | Last Layer |
|---|---|---|---|---|
| **Cosine** | $78.6_{\pm 0.2}$ | $78.3_{\pm 0.1}$ | $78.9_{\pm 0.3}$ | $78.9_{\pm 0.1}$ |
| **Exp Decay Fast** | $77.0_{\pm 0.1}$ | $77.1_{\pm 0.1}$ | $77.3_{\pm 0.2}$ | $76.8_{\pm 0.6}$ |
| **Exp Decay Slow** | $77.7_{\pm 0.3}$ | $77.3_{\pm 0.4}$ | $77.6_{\pm 0.1}$ | $77.5_{\pm 0.3}$ |
| **Linear** | $78.7_{\pm 0.6}$ | $78.4_{\pm 0.3}$ | $78.8_{\pm 0.5}$ | $78.7_{\pm 0.1}$ |
| **None** | $74.7_{\pm 1.1}$ | $77.6_{\pm 0.2}$ | $\mathbf{79.2}_{\pm 0.4}$ | $\mathbf{79.2}_{\pm 0.2}$ |
| **Piecewise Const** | $78.0_{\pm 0.2}$ | $79.1_{\pm 0.5}$ | $79.1_{\pm 0.5}$ | $\mathbf{79.2}_{\pm 0.0}$ |

Table 1: Test accuracy of a VGG network trained on CIFAR10 with various learning rate schedules and Cooling modes. The different Cooling modes perform at least on par with baselines on all LR schedules. Cooling considerably outperforms the baseline when no schedule is used. Significant gains are also achieved when a piecewise constant schedule is used. (All means and standard deviations are computed over three runs.)

Table 2 shows that Cooling works well for both ReLU and CReLU activation functions. In particular, CReLU diverged in all of our experiments without learning rate schedules, but converged when Cooling was used. As for ReLU, we note that last layer and periodically redistributed Cooling outperform pure distributed Cooling.

| Activation | No Cooling | Distributed | Periodic | Last Layer |
|---|---|---|---|---|
| **CReLU** | divergence | $79.7_{\pm 0.4}$ | $84.2_{\pm 0.1}$ | $84.4_{\pm 0.2}$ |
| **ReLU** | $74.7_{\pm 1.1}$ | $77.6_{\pm 0.2}$ | $79.2_{\pm 0.4}$ | $79.2_{\pm 0.2}$ |

Table 2: Test accuracies of a VGG network trained without a learning rate schedule on CIFAR10 with different activation functions and Cooling modes. Last layer scaling and periodically redistributed scaling perform best for both activation functions. Training diverges for CReLU without Cooling and without a LR schedule. (All means and standard deviations are computed over three runs.)

The effect of the Cooling factor on the inverse of the network temperatures is shown in Figure 2. In the left plot, where a smaller Cooling factor and last layer Cooling is used, the inverse temperature grows much more slowly and only exceeds 100 after 19 epochs. In the right plot, where periodically redistributed Cooling with $\tau_{\text{re}} = \tau_{\text{max}} = 100$ is used, the first temperature reset already happens after 9 epochs.

We present further experimental results on CIFAR10 in Appendix B.

### 4.3 Image Classification: CIFAR100

**Setup.** We train a ResNet50 network (He et al., 2016b) on the CIFAR100 dataset (Krizhevsky, 2009). The network consists of a sequence of 50 convolutional layers of varying filter sizes and channel numbers, followed by a global average pooling layer. The network uses skip connections and has approximately 23.5 million trainable parameters in total. The hidden layers either use the ReLU (Fukushima, 1980; Nair & Hinton, 2010) or the CReLU (Shang et al., 2016) activation. In our learning rate schedule ablations, we try out (1) no schedule, (2) a piecewise linear schedule with drops by a factor of 0.1 after 80, 120, 160 and 180 epochs. We train the network for 200 epochs. We use a batch size of 64.

**Results.** Similar to the CIFAR10 experiments, we present an ablation on the effect of different Cooling factors on last layer Cooling. In Figure 1 (right) we notice the same pattern: Cooling factors $\kappa$ that do not exceed 1.0 yield neural network models that outperform the baseline. On the other hand, we observe once more that $\kappa > 1$ leads to the divergence of network training.

### 4.4 Semantic Segmentation: ADE20K Dataset

**Setup.** We train a small U-Net Ronneberger et al. (2015) architecture on the challenging ADE20K dataset Zhou et al. (2019), which includes 150 semantic categories. This dataset contains 20,000

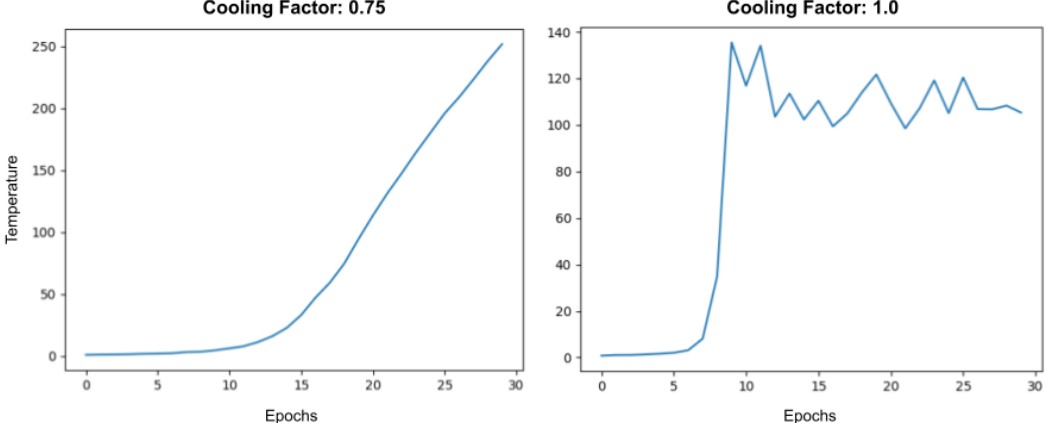

Figure 2: Comparison of the increase of network temperatures for different Cooling factors. Both images show the training of a VGG network on CIFAR10 with ReLU activations. Left: last layer Cooling, CF: 0.75. Right: periodically redistributed Cooling, CF: 1.0. We note that a larger Cooling factor causes a much steeper increase in the temperature. For last layer Cooling, the temperature keeps on growing, whereas for periodically redistributed Cooling, the temperature is redistributed whenever it exceeds a value of 100.

images for training and 2,000 images for validation, on which we report results. We leave aside 320 images from the training set when performing Cooling. We work with images of size $256 \times 256$ and our U-Net architecture has $\approx 9$ million trainable parameters.

**Results.** We compare our proposed last layer Cooling against the baseline, with no temperature scaling. For the former we obtain 22.1 mIoU and 71.9% accuracy, while for the latter we obtain 21.0 mIoU and 70.9% accuracy. This shows that our proposed cooling method is also beneficial on denser, pixel-wise classification tasks. Further investigation on larger architectures and across multiple design choices could further reveal the full potential of Cooling.

## 5 DISCUSSION AND CONCLUSION

Our proposed *Cooling* method to adaptively calibrate classification neural networks produces significant benefits in terms of network performance and training stability. Theoretical and empirical findings point to significant benefits resulting from differently scaled gradients during network training. As a result of experiments on different tasks, datasets and network architectures, as well as an ablation study on different hyperparameter settings we find that that Cooling gives a significant performance benefit over relevant baselines. In particular, we notice that Cooling greatly reduces the need for a learning rate schedule.

Even though all versions of Cooling re-scale the network to the same mathematical function, they all produce differently parameterised networks. This reparameterisation has a strong impact on gradients, resulting in different network functions as training progresses. This raises the question: *What are the general conditions on the parametrisation of a network to achieve optimal training?*

Another highlight of our work is the connection between calibration and the learning rate. There are indications that well-calibrated networks are more stable in training and less reliant on the 'right' learning rate schedule. Since training stability is critical in a number of classification tasks (e.g. when training the discriminator of a GAN), a deeper investigation into the relation between calibration and training stability could be a promising direction for future research.

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
