# OpenReview forum: "How to Keep Cool While Training"
_ICLR.cc/2023/Conference — Submitted to ICLR 2023_

### Official Review · Reviewer_kQKT · 2022-10-21

**Confidence:** 4
**Correctness:** 3
**Technical Novelty And Significance:** 2
**Empirical Novelty And Significance:** 1
**Recommendation:** 3

**Clarity, Quality, Novelty And Reproducibility:**

The paper is relatively clear, and the method is novel. The reviewer has doubts on the reported results and no code is provided.

**Strength And Weaknesses:**

Pros:

1, the analysis of the effects of different cooling strategies on gradients is insightful.

Cons:

**1, Lack of experimental support.** While the method is well developed and studied analytically, the experiments do not fully support the strong claims of the paper. Specifically, classification experiments are only conducted on CIFAR10 and CIFAR100	,¬ both which are considered small datasets by today’s standard. This is especially true when the main claim of the paper is improved performance and stability. Moreover, the accuracy reported for CIFAR10 using VGG seems too low compared to the performance using the same architecture by other open-sourced codes, e.g., [1].

**2, Contribution of distributed cooling and periodic cooling not strong.** A large component of the paper is on distributed cooling and periodic cooling. However, according to Table 1, neither of them is noticeably better than simple last layer cooling, if not worse.

[1] https://github.com/chengyangfu/pytorch-vgg-cifar10


**Summary Of The Paper:**

The paper proposes to incorporate temperature scaling, a popular post-training confidence calibration technique, into training to improve performance and training stability.  Temperature scaling in confidence calibration changes the confidence of a network but does not affect the ordering of predictions. The paper hypothesizes that temperature scaling can regulate gradients if it is used during training.  Specifically, three variants: last layer cooling, distributed cooling and periodic cooling are proposed. For example, last layer cooling uses a temperature parameter in the last linear layer which is calibrated on a held-out set after every training epoch.

**Summary Of The Review:**

The paper aims to show an overlooked benefit of temperature scaling during training. While the analytical analysis is detailed, it lacks enough experiments to support the claims, making the results less convincing.

---

> ### Author Response · Authors · 2022-12-12
> **Reply**
>
> We thank the reviewer for their constructive review. We note that the experiments displayed in the paper are performed using a smaller *VGG-type* network, not one of the networks from the original VGG paper. We have now also run experiments with the bigger VGG-B network on CIFAR10:
>
> **Results with VGG-B on CIFAR10**:
> | BatchNorm | Dropout | LR Schedule      | Cooling Type | Training Accuracy | Test Accuracy |
> | ----------- | ----------- | ----------- | ----------- | ----------- | ----------- |
> | True | True | Cosine      | None       | 99.9% | 88.8% |
> | True | True | Piecewise Constant   | None        | 99.8% | 89.6% |
> | True | True | None | None | 97.8% | 88.0% |
> | True | True | None | Last Layer | 97.9% | 88.2% |
>
> It seems that last layer scaling gives only a marginal benefit over the non-Cooling baseline without learning rate schedule.
>
> **Regarding distributed Cooling and periodic Cooling**
> We agree with the reviewer that if taking the maximum over all cooling factors, distributed and periodic Cooling perform worse, when being compared to last layer Cooling. However, distributed and periodic Cooling are more stable over different cooling factors and they are less reliant on a specific cooling factor (as opposed to last layer Cooling, which often works well for a cooling factor of 0.7, but which works less well for cooling factors larger than 1.0).

---

### Official Review · Reviewer_ZSwR · 2022-10-24

**Confidence:** 3
**Correctness:** 3
**Technical Novelty And Significance:** 2
**Empirical Novelty And Significance:** 3
**Recommendation:** 5

**Clarity, Quality, Novelty And Reproducibility:**

Clarity:
The cooling algorithm’s effect is clearly explained in an aspect of the gradient. In addition, I have a question that vector scaling can be helpful to address the heating in learning, and the precise relationship to calibration is vague to me. In conclusion, the authors stated that there are indications that well-calibrated networks are more stable in training and less reliant on the ‘right’ learning rate schedule. However, there is no clear evidence in the experiments that the cooling algorithm is more calibrated (although it can be expected, it should be clarified). If you can report this, I have a question, can the degree of improvement in calibration cause more stability in learning?

Quality:
Well-written but has some missing in the formulation, such as $\beta^{-}, \beta^{i}$ in Definition 3.2. The formulation is not well-organized and to be understood easily. The motivation is simple in the optimization aspect. However, there are rooms to explore and research, such as vector scaling and intensive experiments.

Novelty & Reproducibility:
The paper provides a somewhat novel algorithm, and it is expected that reproducibility can be possible.


**Strength And Weaknesses:**

Pros:
This paper proposed an algorithm to address the optimization problem in deep networks. The algorithm is simple to be applied in practice, and the simulation results can provide the basic effect of the cooling algorithm.

Cons:
Calibration has its area. Therefore, the authors must show the precise relationship between calibration and cooling in optimization in detail. Also, this paper is more related to optimization topics. Experiments are too limited because only scheduling is examined. Maybe Batch-Normalization Dropout and other regularized methods should be compared.



**Summary Of The Paper:**

This paper proposed the temperature-based cooling algorithm to achieve better performance in optimization. The authors show better gradient scaling and reducing the need for scheduling. The starting point of the cooling algorithm is that miscalibration can worsen network learning. Specifically, the authors consider last-layer cooling, distributed cooling, etc. The gradient analysis is interesting, and experimental results provide the effects of the cooling algorithm compared to naïve approaches.

**Summary Of The Review:**

I’ve not convinced of the relationship between calibration and learning stability in a transparent manner. The paper is
well-written. However, it can have many issues to be solved.

---

> ### Author Response · Authors · 2022-11-14
> **Request for 2 Clarifications**
>
> We thank the reviewer for their thoughtful and constructive review. We will aim to carry out the suggested experiments (involving BatchNorm, Dropout and vector scaling) before the end of the rebuttal period and send a detailed rebuttal thereafter, including replies to all of the reviewer’s questions.
>
> **We would like to ask the reviewer for two clarifications:**
> * We closely examined Definition 3.2 again, but we were not able to find anything missing or incorrect there. (In particular, we could not find any problem related to $\beta^i$ or a negative power of $\beta$, as suggested in the review.) We would thus kindly like to ask the reviewer if they could point out the exact part of the definition which they consider to be flawed.
> * The review says “3. The formulation is not well-organized and to be understood easily.” We tried to take great care of making section 3 well-structured and easily accessible. The paper contains appropriate LaTeX environments for definitions, theorems and the algorithm. Most of the paragraphs are named in order to make it easier for readers to jump between different parts of the paper. We also aimed for utmost clarity in the proofs of our theorems.
> We re-read all of section 3, but we were not able to find anything we could improve in terms of content organisation and understandability. We would therefore kindly like to ask the reviewer if they could point towards concrete issues that they perceive as flaws. We are very happy to fix any unclear points.

---

> ### Author Response · Authors · 2022-12-12
> **Reply**
>
> We thank the reviewer again for their constructive review and for their response to our request for clarifications, which we will take into consideration.
>
> We have performed all the experiments that the reviewer requested:
>
> **Results with VGG and Vector Scaling on CIFAR10**
> Compared to the 79.2% test accuracy achieved by temperature scaling, vector scaling performs significantly worse with a test accuracy of only 74.7%. A potential reason for this underperformance could be that vector scaling has 10 times as many parameters, so it can more easily “overfit” on the validation dataset. Moreover, optimising the single temperature parameter is significantly faster than optimising one parameter for each class. Therefore, we conclude that temperature scaling seems to be the better method, when compared to vector scaling.
>
> ### Experiments with BatchNorm and Dropout
>
> **Results with VGG on CIFAR10**:
> | BatchNorm | Dropout | LR Schedule      | Cooling Type | Training Accuracy | Test Accuracy |
> | ----------- | ----------- | ----------- | ----------- | ----------- | ----------- |
> | False | False | Cosine      | None       | 100.0% | 78.6% |
> | False | False | Piecewise Constant   | None        | 100.0% | 78.0% |
> | False | False | None | None | 95.4% | 74.7% |
> | False | False | None | Last Layer | | 95.4% | 79.2% |
>
> **Results with VGG and BatchNorm on CIFAR10**:
> | BatchNorm | Dropout | LR Schedule      | Cooling Type | Training Accuracy | Test Accuracy |
> | ----------- | ----------- | ----------- | ----------- | ----------- | ----------- |
> | True | False | Cosine      | None       | 100.0% | 81.0% |
> | True | False | Piecewise Constant   | None        | 100.0% | 80.6% |
> | True | False | None | None | 100.0% | 80.3% |
> | True | False | None | Last Layer | 100.0% | 81.3% |
>
> **Results with VGG and Dropout on CIFAR10**:
> | BatchNorm | Dropout | LR Schedule      | Cooling Type | Training Accuracy | Test Accuracy |
> | ----------- | ----------- | ----------- | ----------- | ----------- | ----------- |
> | False | True | Cosine      | None       | 100.0% | 83.1% |
> | False | True | Piecewise Constant   | None        | 100.0% | 82.6% |
> | False | True | None | None | 100.0% | 80.4% |
> | False | True | None | Last Layer | 100.0% | 80.5% |
>
> **Results with VGG and both BatchNorm and Dropout on CIFAR10**:
> | BatchNorm | Dropout | LR Schedule      | Cooling Type | Training Accuracy | Test Accuracy |
> | ----------- | ----------- | ----------- | ----------- | ----------- | ----------- |
> | True | True | Cosine      | None       | 100.0% | 85.2% |
> | True | True | Piecewise Constant   | None        | 100.0% | 85.0% |
> | True | True | None | None | 100.0% | 82.8% |
> | True | True | None | Last Layer | 100.0% | 83.6% |
>
> We conclude that our method works best when using BatchNorm, but it does not outperform the baseline LR schedules when dropout is used. Maybe the problem is that tau is optimised with respect to a network where the weights were not dropped. Therefore, further investigation may be necessary to find an adaptation of our method suited to networks applying dropout.

---

### Official Review · Reviewer_DpjZ · 2022-10-26

**Confidence:** 4
**Correctness:** 3
**Technical Novelty And Significance:** 4
**Empirical Novelty And Significance:** 4
**Recommendation:** 6

**Clarity, Quality, Novelty And Reproducibility:**

The paper is well-written and clear, with good quality and original ideas. The experiments seem reproducible.


**Strength And Weaknesses:**

Strengths:

* The paper is well-written and clear.
* The idea of using calibration during training to improve performance can be considered novel and the methods are well justified and simple.
* The empirical advances are nicely complemented with a theoretical study and discussion about what might be the reasons for the achieved improvements.


Weaknesses:

* My main concern is that in CIFAR10 classification the authors have only considered an architecture that results in accuracies much below 80%, whereas modern architectures (plus data augmentation) go well beyond 90%. It would be important to experiment with at least one better architecture also. In CIFAR100, a better architecture has been used, but also for this dataset there is room for improvement so that the baseline method would be at least 70% accurate.

* In the introduction, the paper promises to measure ECE (expected calibration error) but never gets to doing it, not even in the appendix. On one hand, this promise could simply be dropped, but actually it would be an important thing to do, because calibration is an important part of the motivation and intuition behind the proposed method. For example, it would be good to have plots about how ECE (evaluated on test data) is evolving during training alongside with accuracy improvements.

* Figure 1 (left) starts from the cooling factor value of about $0.5$. In order to support the discussion better, lower values of the cooling factor could have also been considered in that plot.

* It would have been good to see more figures about gradient norm evolution in the appendix, perhaps even for all situations considered in Table 1. This would help to compare whether the evolution achieved with the proposed methods is different or similar to the evolution observed with different learning rate schedules.

Minor weaknesses:

* Some parentheses are missing in the proof of Lemma A.1., e.g. the first formula should be $x'_1=\beta\rho(\beta^{-1}(W'_1 x_0+b'_1))$.

**Summary Of The Paper:**

The paper proposes to use temperature scaling during the training process, thus improving accuracy and reducing the need for learning rate schedules. In the theoretical part of the paper, the gradients have been calculated analytically, comparing the gradients with and without the proposed modifications. In the experimental part, the proposed methods have been applied on several tasks and neural architectures, showing that the methods result in improvements in classification accuracy.

**Summary Of The Review:**

A strong paper with one key problem that the considered architectures are quite far from the state-of-the-art (CIFAR-10 accuracies below 80% as opposed to more than 90%). While in principle, it is good to do experiments with architectures requiring less resources, in this particular case it would be important to know whether the observed accuracy improvements are specific to the considered architectures or whether they also carry over to the state-of-the-art architectures. Also, more information could hav
e been provided in the appendix, in particular about gradient norm evolution as well as calibration evolution.

---

> ### Author Response · Authors · 2022-12-12
> **Reply**
>
> We thank the reviewer for the positive and encouraging feedback and the valuable suggestions for additional experiments, which we have performed in the meantime.
>
> *My main concern is that in CIFAR10 classification the authors have only considered an architecture that results in accuracies much below 80%, whereas modern architectures (plus data augmentation) go well beyond 90%. It would be important to experiment with at least one better architecture also. In CIFAR100, a better architecture has been used, but also for this dataset there is room for improvement so that the baseline method would be at least 70% accurate.*
>
> We have now trained a ResNet50 on CIFAR10 and two more modern architectures (EfficientNet and ConvNeXt) on CIFAR100.
>
> **Results with ResNet50 on CIFAR10**
> | LR Schedule      | Cooling Type | Test Accuracy |
> | ----------- | ----------- | ----------- |
> | Cosine      | None       | 92.0% |
> | Piecewise Constant   | None        | 92.0% |
> | None | None | 90.8% |
> | None | Last Layer | 93.2% |
>
> We see that the performance benefit of using last layer Cooling is twice as big as compared to a learning rate schedule.
>
> **Results with EfficientNet on CIFAR100**
> | LR Schedule      | Cooling Type | Training Accuracy | Test Accuracy |
> | ----------- | ----------- | ----------- | ----------- |
> | Cosine      | None       | 99.7% | 36.0% |
> | Piecewise Constant   | None        | 95.3% | 36.1% |
> | None | None | 94.9% | 37.0% |
> | None | Last Layer | 96.9% | 37.8% |
>
> **Results with ConvNeXt on CIFAR100**
> | LR Schedule      | Cooling Type | Training Accuracy | Test Accuracy |
> | ----------- | ----------- | ----------- | ----------- |
> | Cosine      | None       | 1.1% | 1.0% |
> | Piecewise Constant   | None        | 1.0% | 1.0% |
> | None | None | 100.0% | 33.3% |
> | None | Last Layer | 100.0% | 33.1% |
>
> We observed poor results in general with the more recent network types. We note that starting from a weak baseline, the EfficientNet produces better results with Cooling. For the ConvNext, the networks did not converge with cosine and piecewise constant learning rate schedules. When no learning rate schedule was being used, the baseline results were rather weak and the use of last layer Cooling did not make a difference.
> It seems that a specific hyperparameter configuration may be necessary to achieve better results with the more modern networks on CIFAR100.
> We also note that the fact that the ResNet50 had a baseline test accuracy of 69% (slightly less than the desired baseline test accuracy of 70% desired by the reviewer) does not mean that there is a lot of room for improvement: When searching the literature, we failed to find an architecture whose baseline accuracy is significantly higher, unless the network is pre-trained on a bigger dataset like ImageNet. In fact, all the networks we found that had a higher accuracy on CIFAR100 used transfer learning.
>
> **Regarding plots of the ECEs and gradient norms**
> We thank the reviewer for their valuable suggestions for more plots. We have added additional gradient norm plots in Figures 4-6 on pages 14-16 and the ECE plots in Figures 7-8 on pages 17-18 in the supplementary material.
>
> **Regarding showing additional Cooling factors in Figure 1**
> We thank the reviewer for this suggestion. We will show these Cooling factors in the next version of our paper.
>
> **Missing parentheses in the proof of Lemma A1**
> We commend the reviewer for their detailed reading. We have fixed this mistake in the updated pdf.

---

### Decision · Program_Chairs · 2023-01-20

**Decision:**

Reject

**Justification For Why Not Higher Score:**

After the rebuttal period two out of three reviewers vote for rejection. The third reviewer who voted for a borderline accept also points out that the model  architectures are quite far from the state-of-the-art for important image benchmarks.

**Justification For Why Not Lower Score:**

N/A

**Metareview: Summary, Strengths And Weaknesses:**

The paper proposes to use temperature scaling during neural network training as a means to improve model performance and reducing the need for learning rate schedules.
The novelty of the methods was found to be moderate.
Related methods and different settings for neural network optimization are not sufficiently evaluated. The authors have added additional experiments on  settings for neural network optimization. Yet their additional experiments did not show consistent improvements over the state of the art.
A key problem pointed out by a reviewer is that the considered architectures are quite far from the state-of-the-art on important benchmark data sets. While the authors provided additional experiments on these benchmarks, these still fell short of reaching state of the art.
Minor issues with the mathematical notation where pointed out.